# HICO-GT: Hidden Community Based Tokenized Graph Transformer for Node Classification

## Abstract

Graph Transformers have been proven to be effective for the node classification task, of which tokenized graph Transformer is one of the most powerful approaches. When constructing tokens, existing methods focus on collecting multi-view node information as the Transformer's input. However, if a type of tokens only includes nodes having relations with a target node from one perspective, it will not provide sufficient evidence for predicting unknown labels. Directly applying self-attention to all tokens may also produce contradictory information as they are selected by distinct rules. Meanwhile, as an emerging concept on graphs, hidden communities refer to those with relatively weaker structures and being obscured by stronger ones. In this paper, inspired by the hidden community clustering method, we design a new multi-view graph Transformer called HICO-GT. We first reconstruct the input graph by merging the original topology and attribute information. Through an iterative process of weakening dominant and hidden communities in turn, we obtain two subgraphs both containing node information of topological relation and attributed similarity, and then generate two token sequences correspondingly. Along with another neighborhood sequence produced on the original graph, they are separately fed into the Transformer and fused afterwards to form the final representations. Experimental results on various datasets verify the performance of the proposed model, surpassing existing graph Transformers.

## 1 Introduction

The Transformer architecture (Vaswani et al., 2017) has emerged as a crucial tool across various domains of deep learning. More recently, graph Transformers (GTs) (Dwivedi & Bresson, 2020; Kreuzer et al., 2021) provide a new perspective in solving various classical tasks such as node classification. Given a set of labeled nodes on a graph, the goal of node classification is to predict the unknown labels for other nodes. Early graph Transformers (Dwivedi & Bresson, 2020; Kreuzer et al., 2021; Rampášek et al., 2022) address the task by applying the basic Transformer on the whole input graph and obtaining attention scores among all nodes, which brings large computational cost and badly affects the model's efficiency. To overcome the scalability issue, various tokenized GTs (Zhao et al., 2021; Zhang et al., 2022; Chen et al., 2023) are proposed. Instead of using the whole graph, these models only focus on the sampled information related to the target node and convert them into token sequences as the input of Transformer. This strategy allows the model to be trained in a mini-batch manner (Chen et al., 2023), reducing the computational complexity, while it still ensures adequate information for classifying the unlabeled nodes.

There are mainly two categories of tokens in tokenized GTs: neighborhood token and node token. Neighborhood tokens are able to capture the local structural features around the target node, but it is difficult for them to include some global information such as long-range dependencies. On the contrary, node tokens are more flexible as we can focus on the nodes that share similar properties with the target node no matter where they are located on the graph.

When designing node tokens, the key idea is to collect adequate and comprehensive information by limited elements. A recent work (Fu et al., 2024) points out that a single type of token sequence is insufficient to fully represent the graph and proposes VCR-Graphormer that constructs multi-view tokens in the final sequence. To select node tokens, VCR-Graphormer creates topology-aware

or label-aware super nodes and adds a number of virtual edges to the original graph so that the potential tokens become closer to the target node. However, nodes only carrying useful information in one view are not enough to qualify as a token. The virtual edges with unit weight may not lead to the accurate selection as they do not distinguishing the different importance of local and global information. Moreover, the model mixes all types of tokens before sending them to the Transformer, ignoring the potentially contradictory information learned from self-attention. The length of the final sequence and the number choice of the super nodes both increase the complexity as well. To avoid these disadvantages and make the model more capable, we want to design a multi-view tokenized GT in which each type of tokens, with different meanings from each other, can all capture the graph's information from multiple perspectives, and control the computational cost at the same time.

Community detection is another popular topic in graph data mining, which aims to partition a non-attributed graph into communities and the nodes' connections in the same community are much closer than those from different communities. Based on its original definition, He et al. (2018) find that the communities in a graph may have varying strength, and the communities with relatively weaker structures can be obscured by stronger ones. They correspondingly define the hidden community detection task and design a framework to solve it. By uncovering hidden communities, we can obtain information that would otherwise be difficult to extract on graphs. Although this task have different application scenarios from node classification, we can preprocess the input graph and generate higher-quality tokens with the help of hidden community detection strategy.

In this paper, we propose a new graph Transformer called HICO-GT for node classification. To make all node tokens carry topology and attribute information, we first reconstruct the input attributed graph to a non-attributed graph, where the potential node tokens have relations with the target node of varying degrees and constitute dominant and hidden communities. We use an iterative process to weaken their structures in turn and obtain two subgraphs, where the edge weights are automatically decided to reflect the connection strengths among nodes. Then we perform Personalized PageRank on the subgraphs to construct two corresponding token sequences. Along with another neighborhood token sequence produced on the original graph, all the three sequences are fed into the Transformer architecture separately. Eventually, a novel readout function fuses the Transformer outputs to produce the final node representations for classification. Different to the existing method that mainly utilizes the interactions within the detected clusters after adding virtual nodes and edges, HICO-GT obtains two sets of clusters, and each of them enhances its inner connections by weakening the other's structure on the graph to construct the token sequences, which provides a brand new way for designing tokenized GTs. We test the performance of HICO-GT by extensive experiments and it yields satisfying results on both homophilic and heterophilic datasets.

Our main contributions are summarized as follows:

- We reconstruct a non-attributed graph containing dominant and hidden communities. Inspired by the hidden community detection strategy, we obtain two subgraphs and generate two corresponding token sequences both carrying topological and attributed information.

- We design a new tokenized graph Transformer called HICO-GT, in which two node token sequences and one neighborhood token sequence are separately fed into the standard Transformer architecture and fused by a novel readout function afterwards.

- Comparative experiments on various datasets demonstrate the outstanding performance of HICO-GT for node classification. Ablation study and parameter analysis discuss the effects brought by important modules in the model and different values of key parameters.

## 2 Related Work

### 2.1 Graph Neural Network

For node classification task, early GNNs obey the design of deep neural networks that stack several GNN layers to learn node representations. Some researches (Kipf & Welling, 2017; Abu-El-Haija et al., 2019; Zhu et al., 2020) focus on developing topology-aware neighborhood aggregation. There are also a series of methods leveraging the attention mechanism (Veličković et al., 2018; Bo et al., 2021; Kim & Oh, 2021) to strengthen the aggregation operation by introducing attribute information. Meanwhile, popular techniques in advanced deep neural networks, such as residual

connections, have been adopted to improve the performance of GNN (Chen et al., 2020; Xu et al., 2018). Another strategy of designing GNNs attempts to decouple the aggregation operation and feature transformation operation in each GNN layer (Dong et al., 2021; Chen et al., 2024; He et al., 2022). Hence, a multi-layer GNN can be represented as a combination of a multi-hop neighborhood aggregation module and a feature transformation module. In this way, the training cost could be effectively controlled by removing the non-linear activation operation among GNN layers.

## 2.2 Graph Transformer

Graph Transformers have shown promising performance in node classification. Prior models have developed approaches such as Laplacian eigenvector (Dwivedi & Bresson, 2020; Bo et al., 2023) to encode the graph structural information. Some methods introduce linear attention-based strategies (Wu et al., 2022; 2023) to reduce the computation expense of self-attention. More recent studies transform the graph into independent token sequences as the model input, which effectively controls the training cost. NAGphormer (Chen et al., 2023) utilizes the propagation operation to generate neighborhood-wise token sequences. PolyFormer (Ma et al., 2024) combines the node tokens with a node-wise filter through a tailored attention mechanism, achieving both scalability and expressiveness. These models generate tokens only in one view. VCR-Graphormer (Fu et al., 2024) combines the node-wise and neighborhood-wise tokens to construct a hybrid token sequence. However, the independent selection processes reduce the quality of tokens as they do not carry the multi-view information at the same time. In our model, we merge the topological relation and attributed similarity and generate two token sequences from the new graph. These node tokens all have interactions with the target node from two perspectives and can provide more reference for its label prediction.

## 3 Preliminaries

### 3.1 Problem Definition

Consider an undirected and unweighted graph $\mathcal{G} = (V, E, \mathbf{X})$ with $|V| = n$ nodes and $|E| = m$ edges, where $\mathbf{X} \in \mathbb{R}^{n \times d}$ is the attributed matrix and $d$ is the dimension of the node feature vector. In the adjacency matrix $\mathbf{A} \in \{0, 1\}^{n \times n}$, $\mathbf{A}_{ij} = 1$ denotes that there is an edge between nodes $v_i$ and $v_j$, and $\mathbf{A}_{ij} = 0$ otherwise. The normalized adjacency matrix is calculated as $\hat{\mathbf{A}} = (\mathbf{D} + \mathbf{I})^{-1/2}(\mathbf{A} + \mathbf{I})(\mathbf{D} + \mathbf{I})^{-1/2}$, where $\mathbf{D}$ and $\mathbf{I}$ represent the diagonal degree matrix and the identity matrix, respectively. Each node's one-hot label vector for $c$ classes constitute the label matrix of the graph $\mathbf{Y} \in \mathbb{R}^{n \times c}$. For a node set $\mathcal{V}_l$ with known labels, the goal of node classification task is to predict the labels for the nodes in its complementary set $\mathcal{V}_u = \mathcal{V} - \mathcal{V}_l$.

### 3.2 Modularity and Hidden Community Detection

Modularity (Newman & Girvan, 2004) is a popular metric to evaluate the strength of a set of communities (a partition). For a partition of a non-attributed graph, its modularity is defined as

$$Q = \frac{1}{2m} \sum_{ij} [\mathbf{A}_{ij} - \frac{k_i k_j}{2m}] \delta(i, j), \tag{1}$$

where $k$ is the degree of a node, and $\delta(i, j) = 1$ if node $v_i$ and $v_j$ are in the same community, otherwise $\delta(i, j) = 0$. This metric can also be used for weighted graphs by replacing the numbers of edges and degrees with their corresponding edge weight summation.

He et al. (2018) define that a partition with relatively higher modularity score is considered as the dominant partition, while the other one is the hidden partition. They also propose a framework to detect them, which weakens the structures of discovered communities to make those from the other partition emerge, and uses an iterative process to strength both detection results. ReduceWeight is a method to weaken the community structures. It considers the connections outside a community as background noises, and calculates the interior density $p$ and exterior noise density $q$ of a community $C$ with $n_C$ nodes, i.e.,

$$p = \frac{w_{C_{in}}}{\frac{1}{2}n_C(n_C - 1)}, \quad q = \frac{w_C - 2w_{C_{in}}}{n_C(n - n_C)}, \tag{2}$$

where $w_{C_{in}}$ and $w_C$ denote the weight summation of the edges within $C$ and edges having at least one endpoint inside $C$, respectively. Then, ReduceWeight multiplies the interior edge weights by the ratio of $p$ and $q$, $i.e.$,

$$w_{uv} = w_{uv} \cdot \frac{q}{p}, \ \ (u, v) \in E_C. \tag{3}$$

In this way, the community's density is reduced to the same level as background noises.

### 3.3 Transformer Architecture

As a key component of graph Transformers, the Transformer architecture (Vaswani et al., 2017) contains a number of Transformer layers, and each layer is composed of two modules named multi-head self-attention and feed-forward network. For the input feature matrix $\mathbf{H} \in \mathbb{R}^{n \times d}$, a single-head self-attention module projects it into three subspaces, $i.e.$,

$$\mathbf{Q} = \mathbf{H}\mathbf{W}^Q, \mathbf{K} = \mathbf{H}\mathbf{W}^K, \mathbf{V} = \mathbf{H}\mathbf{W}^V, \tag{4}$$

where $\mathbf{W}^Q \in \mathbb{R}^{d \times d_K}, \mathbf{W}^K \in \mathbb{R}^{d \times d_K}$ and $\mathbf{W}^V \in \mathbb{R}^{d \times d_V}$ are learnable weight matrices. Then, the processed feature matrix is calculated as

$$\mathbf{H}' = \text{softmax}\left(\frac{\mathbf{Q}\mathbf{K}^T}{\sqrt{d_K}}\right)\mathbf{V}. \tag{5}$$

Finally, the learned features from multi-head self-attention are concatenated and fed into the feed forward network to generate the output of the current Transformer layer.

## 4 Methodology

In this section, we present our model HICO-GT. We first introduce graph reconstruction process to merge the information of topology and attribute. Then, inspired by hidden community detection task, we separate two partitions of this non-attributed graph and generate two corresponding token sequences. These sequences, along with another one produced by propagation on the original graph, are separately fed into the Transformer. Finally, a novel readout function fuses all processed features to obtain the ultimate representations for classifying the nodes. The overall framework of HICO-GT is shown in Fig. 1.

### 4.1 Graph Reconstruction

In tokenized GTs, there are typically two types of tokens: neighborhood token and node token. We want to select node tokens resembling the target node in the views of both topology and attribute so that they can provide more reference for node classification. Therefore, we first merge the nodes' original topology and attribute information, and reconstruct the input graph into a new non-attributed one $\mathcal{G}_{\text{new}}$. Specifically, we calculate the cosine similarity of the feature vectors for every node pair on $\mathcal{G}$, $i.e.$,

$$\cos(\theta_{ij}) = \frac{\mathbf{X}_i \cdot \mathbf{X}_j}{||\mathbf{X}_i|| \cdot ||\mathbf{X}_j||} = \frac{\sum_{k=1}^d \mathbf{X}_{ik}\mathbf{X}_{jk}}{\sqrt{\sum_{k=1}^d \mathbf{X}_{ik}^2}\sqrt{\sum_{k=1}^d \mathbf{X}_{jk}^2}}, \ v_i, v_j \in V, \ v_i \neq v_j. \tag{6}$$

Then, we sort all the similarities and pick the highest $m$ scores to create a set $S$ containing their corresponding node pairs. For a node pair in $S$, if there is not an edge connecting two nodes, we add one with the weight equal to the similarity score. If there has already been an edge, we reset its weight to the similarity score plus one. And for a node pair that is not in $S$, we keep its original topological relation, $i.e.$,

$$\mathbf{W}_{ij} = \begin{cases} \mathbf{A}_{ij} + \cos(\theta_{ij}), & (v_i, v_j) \in S, \\ \mathbf{A}_{ij}, & (v_i, v_j) \notin S. \end{cases} \tag{7}$$

In this way, we construct a non-attributed graph $\mathcal{G}_{\text{new}}$ to replace the input graph, and both the topology and attribute information are transformed into topological connections. Then we can perform the following process to generate the token sequences on this graph.

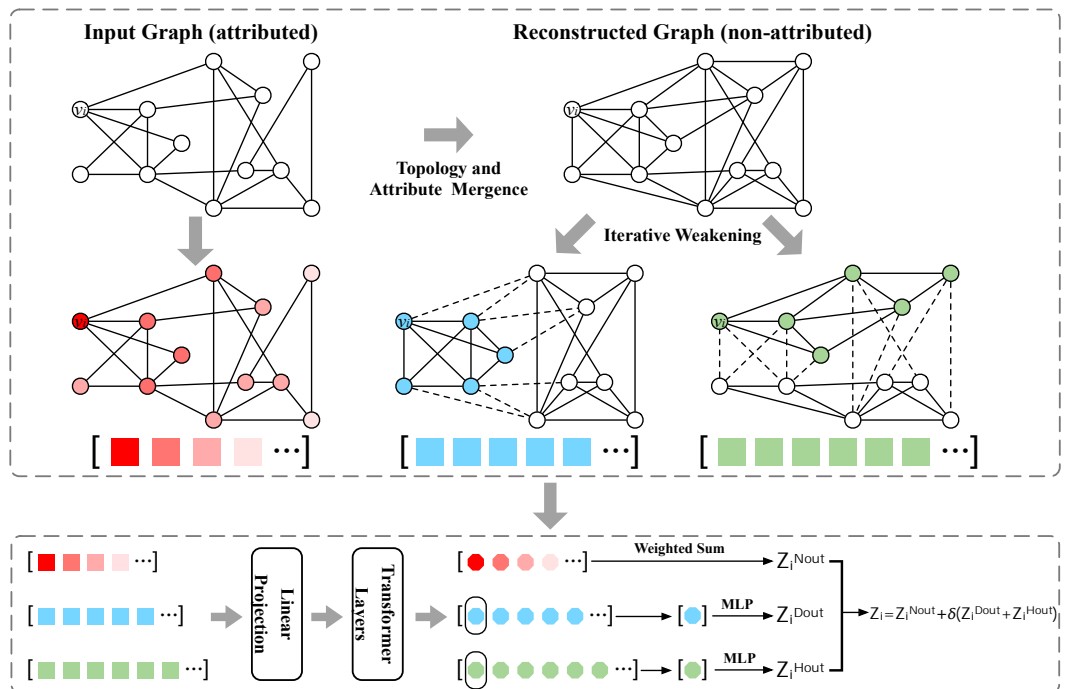

Figure 1: The overall framework of HICO-GT. There are three token sequences in total. The neighborhood sequence is captured on the original input graph. The two series of node tokens are respectively selected on two weakened graphs after an iterative hidden community detection for the reconstructed graph. Weakened edges are represented with dashed lines. The token sequences (squares) are separately fed into a standard Transformer, and the output features (octagons) are fused by a weighted readout function to produce the final node representation.

## 4.2 TOKEN SEQUENCE GENERATION

The task of hidden community detection aims to distinguish the partitions with different cohesiveness and find them separately on a non-attributed graph. On the graph after reconstruction, most of the qualified node tokens tend to constitute communities as they have strong interactions. However, there also exist nodes that share slight topological relation or attributed similarity with the target node but are not included in these communities. Therefore, to dig out all the potential node tokens, we generate a dominant partition and a hidden partition of the graph and construct two corresponding token sequences with the help of the hidden community detection strategy.

Louvain (Blondel et al., 2008) is a classical community detection algorithm based on modularity maximization. For $\mathcal{G}_{\mathrm{new}}$, we first use Louvain to detect one partition, and it naturally resembles the dominant partition as the communities have stronger structures. Once the partition is found, we can calculate the interior and exterior edge density for each community and use ReduceWeight to weaken them. By this means the structures of the hidden communities emerge. Now the community detection operation can uncover more nodes that have relatively weaker connections with the target node. So far we have obtained two partitions but they are not accurate enough. Note that while the communities in the dominant partition cover that in the hidden one, the structures of the latter have an impact on the former as well. Therefore, we also weaken the hidden communities and repeat the above process iteratively to get refined detection results. In the last iteration, we extract the two subgraphs after weakening the two partitions to select the tokens.

As the community weakening process eliminates the distraction from each other, the two subgraphs preserve distinct information, which helps us to select two series of node tokens. Now the automatically-decided edge weights reflect their connection strengths and we use Personalized PageRank (PPR) to compare the relations between all nodes and the target node. For a node $v_i$, PPR

begins with the initial probability vector $\mathbf{r}^{(0)} = \left(\frac{1}{n}, \frac{1}{n}, ..., \frac{1}{n}\right)^\top \in \mathbb{R}^{n \times 1}$ and iterates as:

$$\mathbf{r}^{(k)} = \mu \mathbf{P} \mathbf{r}^{(k-1)} + (1 - \mu)\mathbf{e}, \mathbf{e}_j = \begin{cases} 1, j = i, \\ 0, j \neq i, \end{cases} \tag{8}$$

where $\mathbf{P} = \mathbf{D}^{-\frac{1}{2}} \mathbf{A} \mathbf{D}^{-\frac{1}{2}}$ is the transition matrix and $\mu \in (0, 1)$ is a hyper-parameter to control the probability to restart in each step, and we set $\mu = 0.85$ in our model. Take the dominant nodes as the example, after a certain numbers of steps, PPR gets a probability vector for each node, and we pick the top $Q^D$ highest values to compose a vector $\mathbf{r}_i^D$ in descending order, as well as its corresponding node set $\mathcal{R}_i^D$. Then we concatenate the nodes' features with their PPR values and put them together to form the dominant token sequence. To prevent the loss of information on $v_i$, we set the first token in the sequence to the node itself and assign probability of 1. That is, we obtain $S_i^D = \left\{ \mathbf{X}_i || 1, \mathbf{X}_{\mathcal{R}_{i,1}^D} || \mathbf{r}_{i,1}^D, \mathbf{X}_{\mathcal{R}_{i,2}^D} || \mathbf{r}_{i,2}^D, \cdots, \mathbf{X}_{\mathcal{R}_{i,Q^D}^D} || \mathbf{r}_{i,Q^D}^D \right\}$ for $v_i$. We construct the hidden token sequence $S_i^H$ in the same way. The details of the whole generation process of these two sequences are presented in A.1.

By constructing the above token sequences, we select the nodes sharing similar topology and attribute information and aggregate them to the target node. However, during the graph weakening process, we may lose some local structural information from the target node's neighborhood on the original graph. Therefore, we construct another sequence to collect the neighborhood features. Specifically, we first add a self-loop edge with a weight of 1 for each node on graph $\mathcal{G}$ to pay more attention to the node's own features. Then, the neighborhood aggregation is performed by iteratively multiplying the normalized adjacency matrix with the feature matrix, *i.e.*,

$$\mathbf{X}^{(k)} = \hat{\mathbf{A}} \mathbf{X}^{(k-1)}, \ \ \mathbf{X}^{(0)} = \mathbf{X}. \tag{9}$$

For node $v_i$, we put together its corresponding vectors from the matrices after each step of propagation, forming the local feature tokens from its $K$-hop neighborhood $S_i^N = \left\{ \mathbf{X}_i^{(0)}, \mathbf{X}_i^{(1)}, \cdots, \mathbf{X}_i^{(K)} \right\}$, where $K$ is the maximum steps.

### 4.3 Transformer Backbone and Fusion Operation

After obtaining token sequences from three different perspectives to describe node $v_i$'s features, the next stage is to use the Transformer module to process them. To feed multiple sequences into the Transformer, some existing methods (Fu et al., 2024) choose to combine them into a single sequence, which mixes all types of information and prevents the model from learning self-attention in each view. Moreover, the time complexity of self-attention mechanism is quadratic in the input length (Keles et al., 2023), and thus it incurs a lower cost to process each token sequence separately than putting the multi-view tokens together. Therefore, we apply the former strategy in our model.

We use a similar way to treat the three token sequences for node $v_i$ in the Transformer. Take $S_i^N$ as an example, we first map it to a subspace, *i.e.*,

$$\mathbf{Z}_i^{N,(0)} = \left[ \mathbf{X}_i^{(0)} \mathbf{W}, \mathbf{X}_i^{(1)} \mathbf{W}, \cdots, \mathbf{X}_i^{(K)} \mathbf{W} \right], \tag{10}$$

where $\mathbf{W} \in \mathbb{R}^{d \times d_m}$ is a learnable projection matrix. A standard Transformer layer composes of a multi-head self-attention (MSA) module and a feed-forward network (FFN) module, and they process $\mathbf{Z}_i^{N,0} \in \mathbb{R}^{(K+1) \times d_m}$ by

$$\mathbf{Z}_i^{N,(l')} = \mathrm{MSA}\left(\mathrm{LN}\left(\mathbf{Z}_i^{N,(l)}\right)\right) + \mathbf{Z}_i^{N,(l)}, \tag{11}$$

$$\mathbf{Z}_i^{N,(l+1)} = \mathrm{FFN}\left(\mathrm{LN}\left(\mathbf{Z}_i^{N,(l')}\right)\right) + \mathbf{Z}_i^{N,(l')}, \tag{12}$$

where LN denotes the layer normalization. Through a same $L$-layer Transformer, we obtain the learned representations in three views $\mathbf{Z}_i^{N,(L)}$, $\mathbf{Z}_i^{D,(L)}$ and $\mathbf{Z}_i^{H,(L)}$.

In node token sequences, the elements excluding the first one are all generated mainly from features of other nodes, whose information has already been aggregated into the target node's representation.

Therefore, we abandon them and only send the first elements in $\mathbf{Z}_i^{D,(L)}$ and $\mathbf{Z}_i^{H,(L)}$ into a fully connected layer, *i.e.*,

$$\mathbf{Z}_i^{Dout} = \text{FC}\left(\mathbf{Z}_{i,0}^{D,(L)}\right), \ \ \mathbf{Z}_i^{Hout} = \text{FC}\left(\mathbf{Z}_{i,0}^{H,(L)}\right). \tag{13}$$

As for the neighborhood token sequence, every token, which corresponds to an aggregated feature vector of one single hop in the neighborhood, has a strong correlation with the target node, and thus should be included in the fusion stage. We follow the strategy in (Chen et al., 2023) to acquire the coefficients for each element in $\mathbf{Z}_i^{N,(L)}$, and calculate the weighted sum accordingly.

$$\omega_k = \frac{exp\left(\left(\mathbf{Z}_{i,0}^{N,(L)}||\mathbf{Z}_{i,k}^{N,(L)}\right)\mathbf{W}_1^\top\right)}{\sum_{j=1}^K exp\left(\left(\mathbf{Z}_{i,0}^{N,(L)}||\mathbf{Z}_{i,j}^{N,(L)}\right)\mathbf{W}_1^\top\right)}, \tag{14}$$

$$\mathbf{Z}_i^{Nout} = \mathbf{Z}_{i,0}^{N,(L)} + \sum_{j=1}^K \omega_k \mathbf{Z}_{i,j}^{N,(L)}, \tag{15}$$

where $\mathbf{W}_1 \in \mathcal{R}^{1 \times 2d_m}$ is a learnable projection matrix. Eventually, we fuse the three processed vectors by a summation operation and use a hyper-parameter $\delta$ to control the proportion of the emphasis on node tokens or neighborhood tokens, *i.e.*,

$$\mathbf{Z}_i = \delta(\mathbf{Z}_i^{Dout} + \mathbf{Z}_i^{Hout}) + \mathbf{Z}_i^{Nout}. \tag{16}$$

$\mathbf{Z}_i$ is $v_i$'s final representation and will be fed into the predictor to solve the classification task.

## 5 EXPERIMENTAL RESULTS

### 5.1 DATASETS

We conduct comparative experiments on ten widely used datasets. For homophilic graphs, we select Pubmed, CoraFull, CS and Physics from Deep Graph Library (DGL) [1]. For heterophilic graphs, we select UAI2010 and BlogCatalog from (Wang et al., 2020), as well as Squirrel-filtered, Minesweeper, Tolokers and Questions from (Platonov et al., 2023). We also follow the splitting strategy of the latter four datasets in (Platonov et al., 2023), and apply 60%/20%/20% train/val/test random splits for the others. We use ROC-AUC as the metric to evaluate the models' results for Minesweeper, Tolokers and Questions as there are only two classes for the nodes in these datasets, and use accuracy for the other multi-class tasks. The statistic information of the datasets are summarized in A.2.

### 5.2 BASELINES

We select ten representative methods as the baselines, including GNN-based models, GCN (Kipf & Welling, 2017), GAT (Veličković et al., 2018), APPNP (Klicpera et al., 2019), GPR-GNN (Chien et al., 2021), and GT-based models, NodeFormer (Wu et al., 2022), SGFormer (Wu et al., 2023), Specformer (Bo et al., 2023), NAGphormer (Chen et al., 2023), PolyFormer (Ma et al., 2024), VCR-Graphormer (Fu et al., 2024). The implementation details for HICO-GT and the baselines are introduced in A.3.

### 5.3 PERFORMANCE COMPARISON

We test all models' performance for node classification task on ten datasets, and report the results in Table 1. For the first four homophilic datasets, HICO-GT yields the highest accuracy on almost all the datasets, except for Corafull on which our model is the second best. For the last six heterophilic datasets, HICO-GT outperforms all the GNN-based and GT-based baselines. On heterophilic graphs, neighbor nodes mostly do not share the same label, bringing more challenges for node classification. In HICO-GT, we isolate a specific partition to enhance the hidden community structures and pay more attention to the nodes having relatively weaker connections with the target node, which expands the scope of node token selection. Compared to VCR-Graphormer, another

---

[1] https://www.dgl.ai/

Table 1: Comparison of all models in terms of mean accuracy or ROC-AUC score ± stdev (%). The best results appear in bold.

| Model | Pubmed | Cora. | CS | Physics | UAI. | Blog. | Squi. | Mine. | Tolo. | Ques. |
|---|---|---|---|---|---|---|---|---|---|---|
| GCN | 86.54 ± 0.12 | 61.76 ± 0.14 | 92.92 ± 0.12 | 96.18 ± 0.07 | 74.68 ± 0.82 | 93.56 ± 0.43 | 41.30 ± 0.94 | 72.23 ± 0.56 | 77.22 ± 0.73 | 76.28 ± 0.64 |
| GAT | 86.32 ± 0.16 | 64.47 ± 0.18 | 93.61 ± 0.14 | 96.17 ± 0.08 | 75.17 ± 0.45 | 94.34 ± 0.64 | 35.09 ± 0.70 | 81.39 ± 1.69 | 77.87 ± 1.00 | 74.94 ± 0.56 |
| APPNP | 88.43 ± 0.15 | 65.16 ± 0.28 | 94.49 ± 0.07 | 96.54 ± 0.07 | 76.08 ± 0.66 | 94.21 ± 0.32 | 38.96 ± 0.66 | 88.75 ± 0.74 | 78.36 ± 0.58 | 73.98 ± 0.81 |
| GPR-GNN | 89.34 ± 0.25 | 67.12 ± 0.31 | 95.13 ± 0.09 | 96.85 ± 0.08 | 76.32 ± 0.59 | 95.02 ± 0.34 | 41.09 ± 1.18 | 90.10 ± 0.34 | 77.25 ± 0.61 | 74.36 ± 0.67 |
| NodeFormer | 89.24 ± 0.14 | 61.82 ± 0.25 | 95.68 ± 0.08 | 97.19 ± 0.04 | 73.87 ± 1.39 | 93.33 ± 0.85 | 37.07 ± 9.16 | 86.91 ± 1.02 | 78.34 ± 0.98 | 74.48 ± 1.32 |
| SGFormer | 89.31 ± 0.17 | 65.32 ± 1.08 | 93.62 ± 0.05 | 96.71 ± 0.06 | 75.17 ± 0.49 | 94.32 ± 0.21 | 43.74 ± 2.51 | 77.69 ± 0.96 | 82.07 ± 1.18 | 77.06 ± 1.20 |
| Specformer | 89.19 ± 0.33 | 66.58 ± 0.86 | 96.07 ± 0.10 | 97.30 ± 0.05 | 75.61 ± 0.77 | 96.12 ± 0.23 | 40.20 ± 0.53 | 89.93 ± 0.41 | 80.42 ± 0.55 | 76.49 ± 0.58 |
| NAGphormer | 89.70 ± 0.19 | 71.51 ± 0.13 | 95.75 ± 0.09 | 97.34 ± 0.03 | 78.05 ± 0.75 | 93.88 ± 0.64 | 39.79 ± 0.84 | 88.06 ± 0.43 | 81.57 ± 0.44 | 75.13 ± 0.70 |
| PolyFormer | 90.08 ± 0.14 | 70.81 ± 0.19 | 96.14 ± 0.06 | 97.36 ± 0.03 | 78.63 ± 0.67 | 95.93 ± 0.30 | 42.56 ± 0.96 | 90.69 ± 0.38 | 84.00 ± 0.45 | 77.46 ± 0.65 |
| VCR-G. | 89.77 ± 0.15 | **71.67** ± 0.10 | 95.37 ± 0.04 | 97.34 ± 0.04 | 77.51 ± 0.85 | 93.57 ± 0.42 | 44.44 ± 0.62 | 89.96 ± 0.52 | 82.84 ± 0.60 | 76.03 ± 0.49 |
| HICO-GT | **90.90** ± 0.15 | 71.52 ± 0.12 | **96.20** ± 0.08 | **97.42** ± 0.03 | **79.67** ± 0.54 | **96.15** ± 0.18 | **45.33** ± 0.69 | **90.86** ± 0.32 | **85.27** ± 0.40 | **77.61** ± 0.59 |

Table 2: Comparison with the variants removing two types of tokens. DT and HT stand for dominant token and hidden token, respectively.

| Dataset | HICO-GT | Without DT | Without HT |
|---|---|---|---|
| Pubmed | 90.90 | 90.44 | 90.29 |
| CoraFull | 71.52 | 71.01 | 71.06 |
| CS | 96.20 | 95.69 | 95.61 |
| Physics | 97.42 | 97.12 | 97.09 |
| UAI2010 | 79.67 | 78.05 | 77.89 |
| BlogCatalog | 96.15 | 95.28 | 95.38 |
| Sqirrel-filtered | 45.33 | 42.00 | 41.56 |
| Minesweeper | 90.86 | 90.24 | 90.52 |
| Tolokers | 85.27 | 84.79 | 85.01 |
| Questions | 77.61 | 76.48 | 76.61 |

multi-view tokenized GT, our model merges the topology and attribute information and uses hidden community detection to produce two token sequences instead of independent topology-aware and attribute-aware sequences. Moreover, we send the sequences to the Transformer separately rather than combine them into a single one. The better classification results than VCR-Graphormer confirm the efficacy of our strategy.

## 5.4 ABLATION STUDY

In HICO-GT, we reconstruct a new graph to obtain two partitions, and separately generate two token sequences based on the dominant and hidden community structures, which are the most important modules of the model. To explore how they affect the performance of HICO-GT, we conduct an ablation study that removes each of them from the input of Transformer and rerun the model. The results are shown in Table 2. The performance of each ablation model decrease to varying degrees compared to the intact one. On half the datasets, the model without hidden tokens obtain better

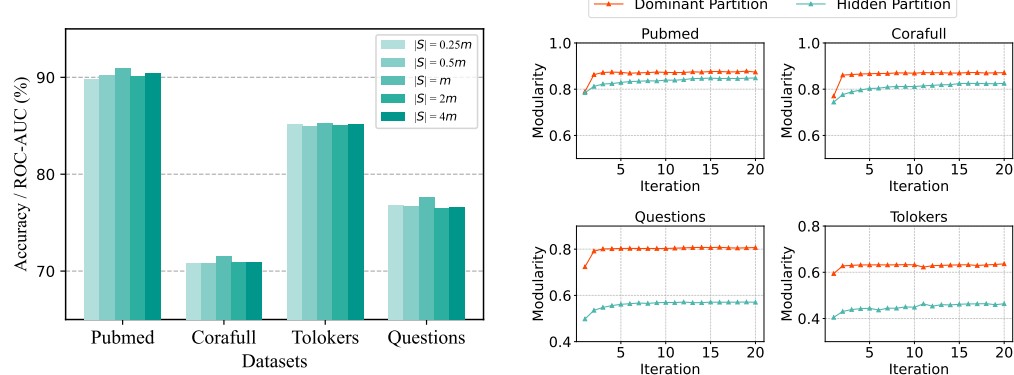

Figure 2: The accuracy / ROC-AUC score of different reconstruction ratio.

Figure 3: The change in modularity value of the two partitions during the iterative process.

results than that without dominant tokens, indicating that although the hidden tokens are selected from relatively weaker community structures, they have the same necessity as the dominant ones.

## 5.5 PARAMETER ANALYSIS

### 5.5.1 THE RECONSTRUCTION RATIO

When reconstructing the non-attributed graph, we create a set $S$ containing a number of node pairs with highest cosine similarity, then add a new edge or reset the edge weight for each node pair. The size of $S$ controls the scale of the new graph, and naturally affects the process of the partition production. We test different values of $|S|$ on two homophilic and two heterophilic datasets, and their corresponding accuracy or ROC-AUC scores are shown in Fig. 2. When $|S| = m$, the model achieves the best results on all the four datasets. A lower value of $|S|$ is insufficient to add adequate attribute information to the new graph, while a higher value may include too many node pairs with negative similarity, resulting in an offset to the original topological relations. Therefore, a set $|S|$ whose size equals to $m$ is created for all datasets.

### 5.5.2 THE MAX ITERATION OF GRAPH WEAKENING

In the operation of token sequence generation, we perform an iterative process to weaken the reconstructed graph and produce two partitions to select node tokens in two views. A higher upper limit of the iteration may lead to more accurate partitions, but it takes greater computational expense at the same time. We run a weakening process of 20 iterations on the same datasets as Fig. 2, and plot the change in modularity in Fig. 3 as a higher modularity value indicates a better partition. On all datasets, the modularity values of the two partitions increase rapidly in the first few iterations. The values of the dominant partition keep basically stable since the 5th iteration, while those of the hidden partition grow a little afterward. To balance the partition quality and the computational cost, we choose the max iteration $T_{max} = 5$ in the experiments.

## 6 CONCLUSION

In this paper, we proposed a new tokenized graph Transformer called HICO-GT for node classification task. We reconstructed a non-attributed graph by merging the topological relations and attributed similarity from the original input graph. Using the strategy of hidden community detection, we produced two weakened subgraphs to separate the information in two views, and select two types of tokens to form the sequences. Another token sequence neighborhood was captured from the input graph. We separately fed all three sequences into the Transformer module and fused them by a weighted readout function to get the final node representation for the classification predictor. Extensive experiments demonstrated the outstanding performance of HICO-GT.

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

# A APPENDIX

## A.1 PSEUDOCODE OF NODE TOKEN SEQUENCE GENERATION

---

**Algorithm 1** Node Token Sequence Generation.

---

**Require:** Reconstructed Graph $\mathcal{G}_{\text{new}}$, target node $v_i$, max iteration $T_{\max}$, size of dominant token sequence $Q^D$, size of hidden token sequence $Q^H$.
**Ensure:** Token sequence $S_i^D$ and $S_i^H$.
1: $C^D, C^H \leftarrow \emptyset$
2: **for** $t = 1 : T_{max}$ **do**
3:    $\mathcal{G}^D \leftarrow \mathcal{G}_{\text{new}}$
4:    **if** $C^H \neq \emptyset$ **then**
5:       $\mathcal{G}^D \leftarrow$ weaken $C^H$ by Eq.(3) on $\mathcal{G}^D$
6:    **end if**
7:    $C^D \leftarrow \text{Louvain}(\mathcal{G}^D)$
8:    $\mathcal{G}^H \leftarrow$ weaken $C^D$ by Eq.(3) on $\mathcal{G}^D$
9:    $C^H \leftarrow \text{Louvain}(\mathcal{G}^H)$
10: **end for**
11: $\mathbf{r}_i^D \leftarrow$ PPR from $v_i$ on $\mathcal{G}^D$, $\mathbf{r}_i^H \leftarrow$ PPR from $v_i$ on $\mathcal{G}^H$
12: sort $\mathbf{r}_i^D$ and $\mathbf{r}_i^H$ in descending order, $\mathcal{R}_i^D$ and $\mathcal{R}_i^H$ are the corresponding node sequences
13: $S_i^D = \left\{ \mathbf{X}_i || 1, \mathbf{X}_{\mathcal{R}_{i,1}^D} || \mathbf{r}_{i,1}^D, \mathbf{X}_{\mathcal{R}_{i,2}^D} || \mathbf{r}_{i,2}^D, \cdots, \mathbf{X}_{\mathcal{R}_{i,Q^D}^D} || \mathbf{r}_{i,Q^D}^D \right\},$

    $S_i^H = \left\{ \mathbf{X}_i || 1, \mathbf{X}_{\mathcal{R}_{i,1}^H} || \mathbf{r}_{i,1}^H, \mathbf{X}_{\mathcal{R}_{i,2}^H} || \mathbf{r}_{i,2}^H, \cdots, \mathbf{X}_{\mathcal{R}_{i,Q^H}^H} || \mathbf{r}_{i,Q^H}^H \right\}$

---

## A.2 STATISTICS ON DATASETS

Table 3: Statistics on datasets.

| Dataset | # Nodes | # Edges | # Features | # Classes |
|---|---|---|---|---|
| PubMed | 19,717 | 88,651 | 500 | 3 |
| CoraFull | 19,793 | 126,842 | 8,710 | 70 |
| CS | 18,333 | 163,788 | 6,805 | 15 |
| Physics | 34,493 | 495,924 | 8,415 | 15 |
| UAI2010 | 3,067 | 28,311 | 4,973 | 19 |
| BlogCatalog | 5,196 | 171,743 | 8,189 | 6 |
| Squirrel-filtered | 2,223 | 93,996 | 2,089 | 5 |
| Minesweeper | 10,000 | 39,402 | 7 | 2 |
| Tolokers | 11,758 | 519,000 | 10 | 2 |
| Questions | 48,921 | 153,540 | 301 | 2 |

## A.3 IMPLEMENTATION DETAILS FOR EXPERIMENTS

We perform hyper-parameter tuning for the baselines by their official implementations. For the model configuration of HICO-GT, we try the the projection dimension in $\{128, 256, 512\}$, the propagation steps in $\{2, 3, \cdots, 6\}$, the number of node tokens in $\{2, \cdots, 20\}$. For the fusion coefficient $\delta$, we first make rough adjustments in $\{0.2, 0.5, 0.8, 1, 1.5, 2, 5\}$, and then make fine-grained adjustments with a granularity of 0.1 in the highest-performance range. Parameters are optimized with AdamW Kingma & Ba (2015) using a learning rate of $\{1e-3, 5e-4, 1e-4\}$ and a weight decay of $\{1e-4, 5e-5, 1e-5\}$. The batch size is set to 2000. The training process is early stopped within 50 epochs. All experiments are performed on a Linux machine with eight NVIDIA RTX 3090 24GB GPUs.

