# OpenReview forum: "HICO-GT: Hidden Community Based Tokenized Graph Transformer for Node Classification"
_ICLR.cc/2026/Conference — Submitted to ICLR 2026_

### Official Review · Reviewer_EzDS · 2025-10-26

**Soundness:** 2
**Presentation:** 2
**Contribution:** 2
**Rating:** 4
**Confidence:** 4

**Summary:**

The proposed methodology addresses the node classification task in non-directed, attributed graphs. The authors present a methodology for performing node classification in graphs through the utilization of Hidden Community and Dominant Community information. The proposed approach first computes the similarity of attributes between nodes using Cosine Similarity, then converts these similarities into edge weights, thereby integrating topology information and attribute information to transform the graph into a non-directed, non-attributed graph. Hidden Communities are discovered in the new graph through Iterative Weakening employing a Reduce Weight technique. To prevent information mixing, the Neighborhood information, Dominant Community information, and Hidden Community information are individually tokenized. The generated tokens are input into a Transformer to produce representations; the Neighborhood Representation is processed in a manner similar to NAGphormer, while the remaining two representations are passed through FC layers, and these are subsequently combined to generate the Final Representation. The generated Final Representation is then input into a predictor to perform classification. The proposed method demonstrates superior node classification performance compared to conventional GNN approaches and other Graph Transformer methods.

**Strengths:**

S1. The application of hidden community detection to tokenized graph transformers is creative and well-motivated. The idea of generating multi-view tokens where each type carries both topological and attributed information is interesting.

S2. Explicitly aims to overcome issues of single-view tokens lacking evidence and potential contradictions from mixing differently derived tokens in prior multi-view GTs. Processing sequences separately is a key design choice.

S3. The model achieves competitive or state-of-the-art results across 10 datasets (4 homophilic, 6 heterophilic), with particularly strong performance on heterophilic graphs where existing methods struggle.

**Weaknesses:**

W1. Computational Complexity Not Fully Addressed. The iterative community detection process (Louvain + weakening × Tmax iterations) adds significant preprocessing cost. No runtime comparisons with baselines provided. Memory overhead of maintaining multiple subgraphs not discussed. The claim of computational efficiency needs empirical validation with timing experiments. In particular, the paper doesn't explicitly analyze the computational complexity or runtime compared to baselines. Given the multiple stages (especially the iterative weakening and multiple PPR runs), the overhead seems potentially substantial, particularly for large graphs, possibly offsetting the benefits gained from tokenization compared to simpler tokenized GTs.

W2. Theoretical justification is limited. Why should hidden communities specifically be useful for node classification? The connection is intuitive but not rigorously established. Why is the weighted sum fusion (Eq. 16) the right way to combine the three token types? Justifications for specific choices within the pipeline (e.g., Louvain vs. other community detection, Reduce Weight vs. other weakening methods, PPR vs. other ranking, the specific fusion formula) could be strengthened.

W3. The graph reconstruction step (Eq. 6-7) combines topology and attributes in a somewhat arbitrary way (cosine similarity + selecting top m pairs). In Section 4.1, the topology information and attribute information of the attributed graph are combined to transform it into a non-attributed graph. In this process, if an edge does not exist, a new edge is generated, and the edge weight is assigned by adding the Cosine Similarity. However, this approach may result in the topology information differing from the original graph, and the values could become excessively magnified compared to existing edges. The authors would benefit from providing an explanation or theoretical justification for this design choice.

W4. Several important details appear to have been omitted. At the end of Section 4, it is stated that three Final Representations are input into the Predictor; however, no description of the Predictor is provided. Additionally, since ROC-AUC and Accuracy employ different loss functions for multi-class datasets and binary classification datasets, respectively, the absence of an explanation regarding how the Predictor was trained makes it challenging to reproduce the results. Furthermore, in Section 5, experimental details such as the number of experimental repetitions are omitted, which raises questions about the statistical significance of the results. It would also be helpful if the authors could clarify whether each token is input into separate L-Layer Transformers or into a single L-Layer Transformer.

**Questions:**

Q1. Could the authors provide an analysis of the computational complexity (e.g., time complexity in terms of nodes/edges) of the HICO-GT pipeline, particularly the graph reconstruction and iterative weakening stages, and compare it empirically (e.g., wall-clock time) to key baselines like VCR-Graphormer or NAGphormer during training and inference?
Q2. What is the actual runtime comparison with baselines? How much overhead does the iterative community detection add?
Q3. Would it be possible for the authors to present results using alternative algorithms (such as Leiden) in addition t the Louvain algorithm?
Q4. Can you provide theoretical or empirical evidence that hidden communities specifically help with node classification?
Q5. How sensitive is the model to the Louvain algorithm's non-determinism?
Q6. Why not learn the fusion weights (δ) rather than tuning them as hyperparameters?
Q7. How do the learned token representations differ between dominant and hidden sequences? Can you visualize this?
Q8. How sensitive is the model's performance to the lengths of the token sequences ($Q^D, Q^H, K$)? Is there a trade-off between performance and sequence length (computational cost)?

---

### Official Review · Reviewer_txmc · 2025-10-29

**Soundness:** 2
**Presentation:** 1
**Contribution:** 3
**Rating:** 4
**Confidence:** 4

**Summary:**

This work focuses on node classification, introduces a frame work to make all node tokens carrying both topology and attribute information. To be specific, it conducts hidden community mining methods on the node attributes similarity graph to identify two hidden important communities, and then use message passing on all the community graphs and original graphs to get embeddings. Finally, they are fed into a Transformer layer following a weighed readout layer to get final embedding for node classification.



The experiments show its competitive results to other graph transformers on ten datasets.

**Strengths:**

1. The experimental results are significant, which achieves the SOTA performance on 9 datasets.
2. The idea that conducts hidden community mining method with reduced weight method are well motivated and designed.
3. The problem and notations are well defined.

**Weaknesses:**

1. Although the hidden community mining method is completely introduced, there lacks analysis on the hidden community mining results. Is the proposed communities intuitively correct? what specific information has it mined from original graph? The authors should analyze/visualize the results more deeply.
2. In the sequence generation part, the motivation to use PPR is not stated.
3. The presentation in the method part is kind of redundant for me to follow. It should be more concise and direct.
4. The framework figure (Figure 1) should include more important details, especially that related to the main idea and contributions.
5. The ablation study is not well disentangled. The authors should include more ablations. For example, only use the original graph (ie, without DT and HT); compare the readout method; compare the PPR and others; etc.
6. Parameter Sensitive analysis is lost (for example, the weighting parameter in Eq. 16).
7. Some other issues about presentations:
   1. Line 46: "There are mainly two categories of tokens in tokenized GTs:" Add references. No literature to support this taxonomy.
   2. Line 84: What is weakening others' structure? it is not clear
   3. Grammar issue: By this means the structures of the hidden communities emerge
   4. Regarding the input of MLP stated in Line 324-325, I do not agree with the reason explained in Lin 322-323: "In node token sequences, the elements excluding the first one are all generated mainly from features of other nodes, whose information has already been aggregated into the target node’s representation." Why does it abandon the information that has already been aggregated from the FC layer? This operation in Eq (13) deserves more intuitive or analytic discussions.

**Questions:**

Please try to address the questions in Weaknesses.

---

### Official Review · Reviewer_nPtT · 2025-11-01

**Soundness:** 2
**Presentation:** 2
**Contribution:** 2
**Rating:** 4
**Confidence:** 4

**Summary:**

This paper presents HICO-GT, a tokenized graph Transformer for node classification that leverages dominant and hidden community structures. Node token sequences and neighborhood information are processed through separate Transformer modules and fused for final representations. Experiments on ten datasets demonstrate competitive or superior performance compared to GNN and graph Transformer baselines.

**Strengths:**

1. The paper is clearly written and easy to follow.

2. The experiments are conducted on multiple graph datasets, including both homophilic and heterophilic graphs. The ablation study in Table 2 demonstrates the necessity of both dominant and hidden tokens.

**Weaknesses:**

1. From Table 1, except for Blog. and Tolo., the proposed method achieves performance comparable to existing works across the ten datasets.

2. Additional comparisons with related GNN and graph Transformer methods [1–2] are needed to further validate the effectiveness of the proposed approach.

3. To more comprehensively demonstrate its effectiveness, the proposed method should also be evaluated on large-scale or long-range datasets.

[1] Luo, Yuankai, Lei Shi, and Xiao-Ming Wu. "Classic gnns are strong baselines: Reassessing gnns for node classification." NeurIPS 2024.

[2] Chen, Jinsong, et al. "Rethinking tokenized graph transformers for node classification." arXiv preprint arXiv:2502.08101 (2025).

**Questions:**

How does the choice of community detection algorithm (Louvain) and weakening schedule (Eq. (3)) affect classification performance compared to alternative algorithms? Have alternatives been attempted?

---

### Official Review · Reviewer_8rpR · 2025-11-01

**Soundness:** 2
**Presentation:** 3
**Contribution:** 1
**Rating:** 2
**Confidence:** 3

**Summary:**

This paper proposes a Hidden Community-based Tokenized Graph Transformer model, named HICO-GT, to address the node classification problem. HICO-GT constructs a new weighted graph by fusing the topological and attribute information of the input graph, and generates two types of node token sequences from this weighted graph via a hidden community detection strategy to address insufficient neighborhood token sequence information. Experimental results verify the model’s effectiveness.

**Strengths:**

1. Community is an inherent property of graphs, and it is reasonable to construct node token sequences using community information.

2. The paper is well organized and explains the method very clearly.

**Weaknesses:**

1. In Section 4.1, it is necessary to calculate the cosine similarity between node pairs and sort these similarity scores, which is an $O(n^2)$ operation.

2. In Section 4.2, PageRank needs to be run on two different subgraphs for each target node, resulting in considerable computational overhead.

3. In each iteration of the model, the Louvain algorithm must be executed on the reconstructed graph. Since the reconstructed graph may be dense, this increases the cost of community detection.

4. The authors do not provide a computational complexity analysis of HICO-GT in the paper.

5. This paper lacks the latest comparative baselines, such as [1] and [2].

[1] Xu X, Zhou Y, Xiang H, et al. NLGT: Neighborhood-based and Label-enhanced Graph Transformer Framework for Node Classification. AAAI, 2025

[2] Zhuo J, Liu Y, Lu Y, et al. Dualformer: Dual graph transformer. ICLR, 2025.

**Questions:**

1. The primary motivation for tokenized graph Transformers is to overcome the quadratic computational complexity between node pairs, thereby achieving scalability. However, the cosine similarity score between pairs of nodes in graph reconstruction can be viewed as a special kind of attention score, which seems to contradict the design intent of tokenized graph Transformers.

2. The cosine similarity has a value range of -1 to 1, so the reconstructed graph may contain negative edge values. However, two classic methods, Louvain and PageRank, both assume non-negative edge weights. Please discuss how to address the special case of negative edges in the reconstructed graph.

---

### Meta-Review · Area_Chair_4SY3 · 2026-01-06

**Summary:**

Overall, reviewers find the idea of leveraging hidden community tokens interesting and the paper clearly written, but they judge the contribution as not yet at the ICLR acceptance bar given substantial concerns about efficiency and experimental coverage; So I recommend reject.

**Reviewer Concerns:**

The main unresolved issues are the heavy and unquantified preprocessing or runtime overhead, missing up-to-date baselines, and insufficient analysis/ablation and reproducibility details (e.g., predictor/training details, seeds/repetitions and sensitivity studies).

**Reviewer Scores:**

The score pattern remains below threshold (one clear reject at 2 and three marginally-below-threshold 4s), and I do not see enough evidence from the reviews/discussion that these core concerns were resolved to justify an upgrade.

---

### Decision · Program_Chairs · 2026-01-26

Reject